# Emerging variants of SARS-CoV-2 NSP10 highlight strong functional conservation of its binding to two non-structural proteins, NSP14 and NSP16

Huan Wang[1], Syed RA Rizvi[1], Danni Dong[1], Jiaqi Lou[1], Qian Wang[1], Watanyoo Sopipong[1], Yufeng Su[2], Fares Najar[3], Pratul K Agarwal[3,4], Frank Kozielski[1], Shozeb Haider[1,5]*

[1]Department of Pharmaceutical and Biological Chemistry, School of Pharmacy, University College London, London, United Kingdom; [2]College of Engineering, Design and Physical Sciences, Brunel University London, Uxbridge, United Kingdom; [3]High-Performance Computing Center, Oklahoma State University, Stillwater, United States; [4]Department of Physiological Sciences, Oklahoma State University, Stillwater, United States; [5]UCL Centre for Advanced Research Computing, University College London, London, United Kingdom

*For correspondence:
shozeb.haider@ucl.ac.uk

**Abstract** The coronavirus SARS-CoV-2 protects its RNA from being recognized by host immune responses by methylation of its 5' end, also known as capping. This process is carried out by two enzymes, non-structural protein 16 (NSP16) containing 2'-O-methyltransferase and NSP14 through its N7 methyltransferase activity, which are essential for the replication of the viral genome as well as evading the host's innate immunity. NSP10 acts as a crucial cofactor and stimulator of NSP14 and NSP16. To further understand the role of NSP10, we carried out a comprehensive analysis of >13 million globally collected whole-genome sequences (WGS) of SARS-CoV-2 obtained from the Global Initiative Sharing All Influenza Data (GISAID) and compared it with the reference genome Wuhan/WIV04/2019 to identify all currently known variants in NSP10. T12I, T102I, and A104V in NSP10 have been identified as the three most frequent variants and characterized using X-ray crystallography, biophysical assays, and enhanced sampling simulations. In contrast to other proteins such as spike and NSP6, NSP10 is significantly less prone to mutation due to its crucial role in replication. The functional effects of the variants were examined for their impact on the binding affinity and stability of both NSP14-NSP10 and NSP16-NSP10 complexes. These results highlight the limited changes induced by variant evolution in NSP10 and reflect on the critical roles NSP10 plays during the SARS-CoV-2 life cycle. These results also indicate that there is limited capacity for the virus to overcome inhibitors targeting NSP10 via the generation of variants in inhibitor binding pockets.

## eLife assessment

This study presents an **important** discovery that the RNA synthesis protein of SARS-CoV-2, the virus that is responsible for COVID 19, has fewer mutations and causes limited conformational changes. The evidence supporting the claims is **convincing**, with robust sequence alignment studies, state-of-the-art protein-protein interaction analysis, and molecular conformational analysis. This work has implications for drug design and will be of broad interest to the general biophysics and structural biology community.

## Introduction

Severe Acute Respiratory Syndrome Coronavirus 2 (SARS-CoV-2), the causative agent for Corona Virus Disease 19 (COVID-19) has infected more than half a billion people and has killed more than 6.4 million by the end of 2022. Despite the development of various vaccines, there is an urgent need to discover and develop effective means of treatment to support the fight against reemerging virus variants. In the past 2 years, extensive research has been carried out to study the molecular determinants of COVID-19 and explore potential therapies. Two specific SARS-CoV-2 targeting drugs and several monoclonal antibodies have recently been approved, such as Paxlovid targeting the main protease (NSP5) (*Owen et al., 2021*) and Remdesivir which targets viral RNA-dependent RNA polymerase (*Sheahan et al., 2020*).

SARS-CoV-2 has an enveloped, positive-sense, single-stranded genomic RNA of ca. 30 kb (*Wu et al., 2020*). There is a 79.5% identity between SARS-CoV-2 and SARS-CoV genome sequences (*Zhou et al., 2020*). The difference in their genomes provided new insights and SARS-CoV-2 is therefore considered a new type of human-infecting β-coronavirus (*Lu et al., 2020*). Understanding the SARS-CoV-2 life cycle and identifying as well as characterizing possible drug targets permits the development of potential therapeutics.

The SARS-CoV-2 genome has 14 open reading frames (ORFs)(*Wu et al., 2020*). The two overlapping ORFs (ORF1a and ORF1ab) are the two critical transcriptional units coding for polyprotein 1 a (PP1a) and polyprotein 1ab (PP1ab), respectively (*Malone et al., 2022*). After cleavage, 16 mature proteins are produced from PP1a and PP1ab, known as nonstructural proteins 1–16 (NSP), and these proteins are responsible for generating the Replication-Transcription Complex (RTC).

After entering the host cell, an essential step is to protect the viral RNA from degradation by the host's innate immunity (*Park et al., 2022*). Similar to the 5′ end of eukaryotic mRNAs, it is essential for viral mRNAs to possess a cap structure that helps the virus evade the immune surveillance of the host (*Ramanathan et al., 2016*). Without a 5′ cap, viral RNA will immediately be degraded in the

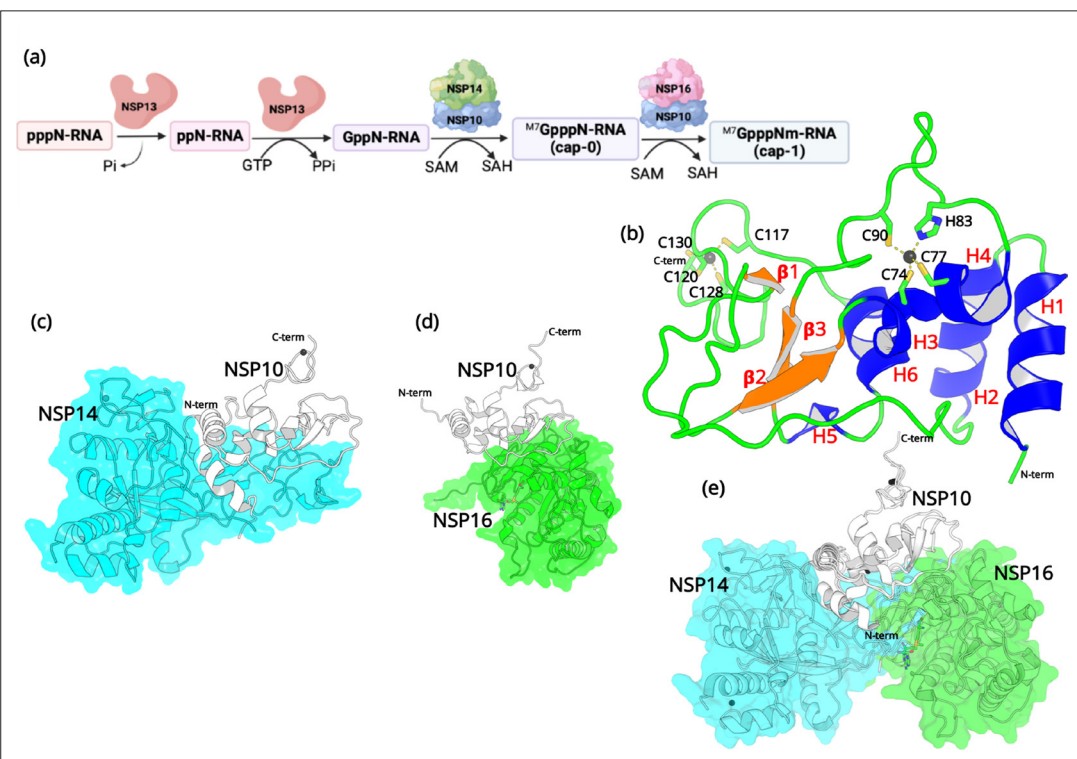

**Figure 1.** Structural features of NSP10. (**a**) The capping mechanism of viral mRNA in SARS-CoV-2 involves four proteins: NSP13 (helicase, orange), NSP14 (green) and NSP16 (pink), and NSP10 (blue). NSP10 acts as a cofactor for both NSP14 and NSP16. (**b**) The structure of NSP10 (PDB entry 6CZT). NSP10 (white/gray cartoon) in complex with (**c**) NSP14 (cyan) and (**d**) NSP16 (green). (**e**) Overlay of NSP10 interacting with NSP14 and NSP16 displaying the partially overlapping interface of both enzymes with NSP10.

cytoplasm (*Viswanathan et al., 2020*). Among the non-structural proteins, both NSP14 and NSP16 are essential for viral RNA capping (*Park et al., 2022*). RNA cap formation in coronaviruses involves various protein-protein and protein-RNA interactions (*Figure 1*).

First, a bifunctional NSP13 (helicase) uses its RNA/NTP triphosphatase (TPase) activity to hydrolyze 5'γ-phosphate of the nascent RNA chain (pppN-RNA), which results in the formation of the diphosphate 5'-ppN end in the ppN-RNA (*Park et al., 2022*). Next, the RNA guanylyltransferase (GTase), also present in NSP13, adds a GMP molecule to generate GpppN-RNA. Subsequently, a cap-0 (m7GpppN-RNA) is formed through N7-methylation of guanylate by the guanine-N7 methyltransferase (N7-MTase) domain of NSP14. Finally, a cap-1 structure (m7-GpppNm-RNA) is generated by further methylation on the 2'-O-position of the first transcribed nucleotide, which relies on the ribose 2'-O-methyltransferases (2'-O-MTase) activity present in NSP16 (*Wilamowski et al., 2021*). During this process, both NSP14 and NSP16 act as S-adenosyl-L-Methionine (SAM) dependent methyl transferases, and SAM acts as the methyl group donor resulting in the formation of the by-product SAH (*Krafcikova et al., 2020*). It is important to note that the proofreading 3'-to-5'exoribonuclease (ExoN) in NSP14 and the 2'-O-MTase in NSP16 require an essential cofactor, NSP10, that acts as an activator and stimulator of enzymatic activities (*Bouvet et al., 2014*). Since the RNA capping mechanism in coronaviruses is necessary for the integrity and stability of viral RNA, the roles that NSP14, NSP16, and more importantly NSP10 are crucial, making them potential targets for antiviral drug design (*Ferron et al., 2012*).

Although NSP10 does not stimulate the N7-MTase activity of NSP14, the ExoN activity of NSP14 relies on the stimulatory factor NSP10 (*Riccio et al., 2022*). The N-terminal ExoN domain of NSP14 is responsible for viral RNA proofreading to maintain a low mutation rate and the integrity of the viral genome (*Lin et al., 2021*). Failure of NSP10 to bind to NSP14 decreases the ExoN activity and thus leads to a significantly lower replication fidelity and high lethal mutagenesis (*Riccio et al., 2022*). In addition, the ExoN activity is also responsible for other viral replication processes such as regulating genome recombination in SARS-CoV-2 (16) and escape from the host immune response (*Becares et al., 2016*).

RNA viruses including coronaviruses such as SARS-CoV-2 have a higher mutation rate than DNA viruses (*Sanjuán et al., 2010*), although the rate in coronaviruses is attenuated compared to other RNA viruses due to the presence of the ExoN activity (*Callaway, 2020*). This ExoN domain is located in the N-terminal part of NSP14 and is capable of correcting errors during RNA replication (*Tahir, 2021*). Mutations rates of $6.677 \times 10^{-4}$ substitutions per site per year have recently been reported for SARS-CoV-2 (*Wang et al., 2022*). Others reported two single-letter mutations per month per SARS-CoV-2 virus (*Callaway, 2020*). This natural evolution process leads to SARS-CoV-2 virus particles containing mutations (also called variants) scattered throughout their genome, which may have advantages for survival and reproduction, or these mutations may be neutral or have a negative impact on survival of the virus (*Sun et al., 2021*). The study of these variations through full genome sequencing has become an essential surveillance tool, as novel SARS-CoV-2 variants of concern (VoC) have emerged over the last 2 years, which display higher infectivity, cause more severe disease, and can escape the effects of vaccination (*Harvey et al., 2021*). Spike, one of the four structural proteins, is one particular example, for which variants emerged that display a higher affinity for the human host cell receptor Angiotensin-converting enzyme 2 (ACE2), explaining an increased impact on transmissibility and immunity for some variants of concern such as Omicron BA.2, BA.4, and BA.5 (https://www.ecdc.europa.eu/en/covid-19/variants-concern). There is clearly a need to monitor novel variants not only for structural but also for non-structural proteins and associated coronavirus proteins (*Davidova et al., 2022*).

In this study, we report a comprehensive analysis of globally collected whole genome sequences (WGS) of SARS-CoV-2 from the Global Initiative Sharing All Influenza Data (GISAID) database. The sequences were aligned to the reference Wuhan/WIV04/2019 genome to identify the entire set of non-synonymous mutations in NSP10 including the ones with the highest frequency of occurrence. We then solved the crystal structure of the most frequent variant of NSP10 to 2.2 Å. The stability and binding affinities for NSP14 and NSP16 of the top three most frequently occurring NSP10 variants were determined using thermal shift (TSA) and microscale thermophoresis (MST) assays. Well-tempered metadynamics simulations were run to sample the free energy landscape of the variants. Our results highlight that NSP10 is more resistant to genetic variations than other SARS-CoV-2 non-structural proteins and that the presence of mutations does not cause any significant structural or

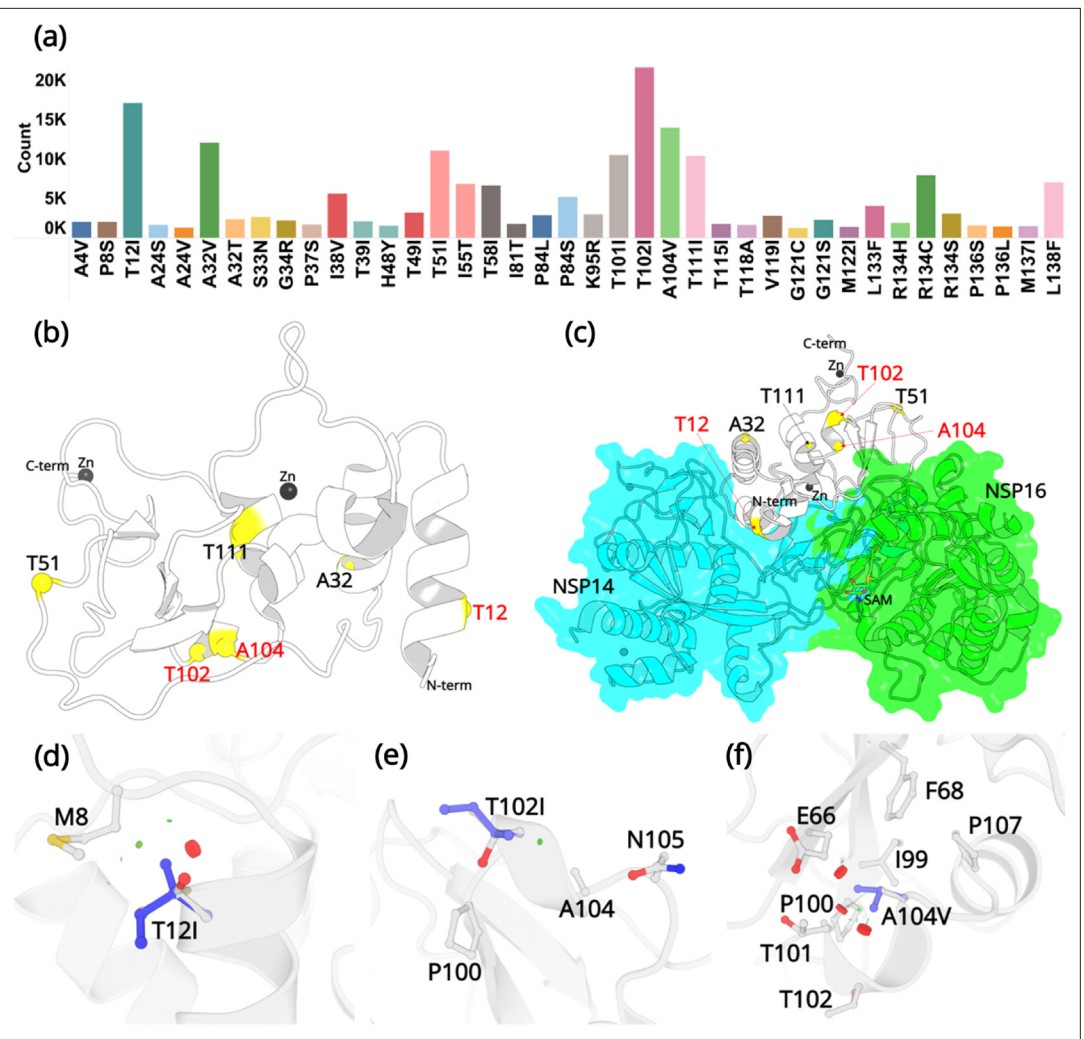

**Figure 2.** Mutations in NSP10. (**a**) Mutation count of the top 39 mutations in NSP10 extracted from >13 million WGS. (**b**) Locations of top six mutations on the NSP10 structure. Mutation positions labeled in red are the most frequently occurring and were used in crystallographic studies. (**c**) The spatial position of the mutations relative to NSP14 and NSP16 structures. The local structural environment of (**d**) T12I, (**e**) T102I, and (**f**) A104V mutations (blue sticks) superimposed on the wild-type structure (white). The red discs represent regions where pairwise overlap of van der Waal radii occurs between the side chain atoms of the point mutation and the surrounding structural elements. The green discs represent regions where atoms are almost in contact.

The online version of this article includes the following source data and figure supplement(s) for figure 2:

**Source data 1.** A detailed table listing all 820 mutations extracted from 7,070,539 sequences.

**Source data 2.** A detailed table listing all 878 mutations extracted from 13,032,424 sequences.

**Source data 3.** Summary of the most frequent variants occurring in NSP10 calculated using Dynamut2.

**Figure supplement 1.** Mutation count of all 820 mutations in NSP10 extracted from 7,070,539 sequences, arranged from residue 1 to 139.

**Figure supplement 2.** Mutation count of all 878 mutations in NSP10 extracted from 13,032,424 sequences, arranged from residue 1 to 139.

**Figure supplement 3.** Protocol used for whole genome sequence analysis.

dynamic change in NSP10. Equally, the most frequent mutations do not have a significant impact on the binding affinity of NSP10 for two of its main interaction partners, NSP14 and NSP16. The effects of these naturally occurring mutations reflect the evolutionary relationship between structurally conserved essential cofactors, their function, and the role they play in the survival of the virus.

## Results

## NSP10 residues have a low frequency of mutation

The SARS-CoV-2 NSP10 sequence analysis was carried out in two phases. All sequences were obtained from the Global Initiative Sharing All Influenza Data (GISAID) database. In the first phase (January 30, 2022), after deleting the sequences with unknown residues (see Materials and methods) in NSP10, 7,070,539 WGS were used for sequence alignment analysis. We concentrated on non-synonymous mutations, those that change one amino acid into another, but ignored synonymous mutations. From these, 1747 unique mutations were identified at different specific amino acids as summarized in (*Figure 2—figure supplement 1*). This is in stark contrast to SARS-CoV-2 NSP1, for which 933 mutations have been identified in only 295,000 genome sequences (*Mou et al., 2021*). The naturally occurring mutations are distributed across the entire NSP10 structure and each and every residue has been mutated, albeit occurring at strikingly different frequencies. A summary table listing all occurring mutations and their frequencies is also provided (*Figure 2—source data 1*). The top three mutations that were identified were at amino acid positions T12, T102, and V104. The NSP10 variants at this position were sent for crystallographic structure determination.

Since our first analysis in January 2022, the SARS-CoV-2 virus has constantly been evolving with multiple variants being reported across the world. Just before the submission of this work, we updated our sequence analysis. A total of 13,032,424 sequences were downloaded and reanalyzed (accessed on December 14, 2022). From these, 2564 unique sequences were identified and the % frequency for the highest occurring mutations was reported (*Figure 2—figure supplement 2*). The updated summary table (*Figure 2—source data 2*) listing all occurring mutations and their frequencies is provided. To our surprise, T12I, T102I, and A104V were still the most frequently occurring mutations, with T102I displaying the highest mutation count.

Twenty-four of the 139 residues display a very low count of less than 100 from more than 13 million genome sequences. For example, the residues forming the two zinc finger domains in NSP10 through association with a zinc ion, fall into this category. The two zinc fingers are thought to provide structural stability to NSP10 although their detailed function remains elusive. Our analysis highlights the importance of the conservation of residues like cysteines and histidines in zinc finger motifs. Although these residues were mutated to various other amino acids, their low frequency of occurrence (below 50), indicates that the mutations at these positions are not favorable for the adaptation or evolution of the virus. The majority of residues, 62, display a medium propensity to mutate with a frequency between 100 and 1000. We then focused on variants with high frequency. In total, 53 mutations occurred more than 1000 times. From these, 39 are considered the most common mutations due to their high frequency compared to the rest of the mutations (*Figure 2a*) and their % frequency exceeded 0.1% (*Table 1*).

We next assessed the impact of the variants on the stability of NSP10 by analyzing the change in free energy ($\Delta\Delta G^{Stability}$) compared to wild-type NSP10. The DynaMut2 algorithm combines Normal Mode Analysis (NMA) methods to capture protein motion and graph-based signatures to represent the wild-type environment to investigate the effects of single and multiple point mutations on protein stability and dynamics (*Rodrigues et al., 2021*). A negative value of $\Delta\Delta G$ refers to protein destabilization by the variant, whereas a positive value implies protein stabilization (*Rodrigues et al., 2021*). Ten mutations including A4V, P8S, L133F, R134C, R134H, R134S, P136L, P136S, M137I, and L138F

**Table 1.** Summary of the top three variants occurring in NSP10 calculated using Dynamut2.
The percent frequency and effect on calculated protein stability are shown. A summary of the top 39 mutations has been listed in *Figure 2—source data 3*. Only mutations with a count larger than 1000 and a percent frequency larger than 0.01% have been included.

| Mutation | Count | %Frequency | $\Delta\Delta G^{Stability}$ | Predicted effect on NSP10 stability | NSP10-NSP14 | | NSP10-NSP16 | |
|---|---|---|---|---|---|---|---|---|
| | | | | | Interface | Deleterious | Interface | Deleterious |
| T102I | 21615 | 0.166 | −0.11 | destabilizing | N | N | N | N |
| T12I | 17120 | 0.131 | −0.68 | destabilizing | Y | N | N | N |
| A104V | 13988 | 0.107 | −0.1 | destabilizing | N | N | N | N |

were not considered as they were missing in the crystal structure. The missing residues are located at the N- and C-termini of NSP10, which suggests that the effects of these mutations may be negligible on the stability of NSP10. The percent frequency and the predicted effects on the stability of the top three NSP10 variants are listed in *Table 1*. A detailed list of all 39 variants is listed in *Figure 2—source data 3*.

The three most commonly identified mutations are T12I, T102I, and A104V. From our analysis of over 13 million sequences, they were identified 17120, 21615, and 13988 times with a % frequency of 0.13, 0.16, and 0.10, respectively. This is in stark contrast to some other, structural and non-structural proteins like the spike protein, E-protein, and the M-protein, which we have been monitoring in real time and are mutating at a much more rapid rate (*Davidova et al., 2022*). Residue T12 is positioned on the helix H1. In the native NSP10 structure, the hydroxyl group side chain is solvent exposed. However, when in complex with NSP14, T12 is present at the interface and makes hydrogen bonding interactions with N61 in NSP14. Mutation to a hydrophobic side chain such as Ile, results in a loss of this interaction. T102 and A104 are present on the proximal and the distal ends of the short helix H5. The side chains of T102 and A104 are also solvent exposed, but far away from the NSP14 or NSP16 protein-protein binding interface. The top three mutations were predicted to have a destabilizing effect on the stability of NSP10, however, they were also predicted to not be deleterious to the structural viability of the protein.

In the native NSP10 structure, the methyl group in the side chain of T12 (helix H1) makes hydrophobic interactions with the M8 side chain. Mutation to the bulkier Ile side chain introduces steric clashes with the N-terminal end of H1. The predicted ΔΔG is –0.68 kcal/mol. The hydroxyl group in the side chain of T102 (helix H5) is solvent exposed and makes interactions with a water molecule in the wild-type NSP10 structure. This interaction is lost in the T102I variant. The predicted ΔΔG is –0.1 kcal/mol. Similarly, the side chain of A104 (Helix H5) is surrounded by the hydrophobic side chains of F68 and I99, while also interacting with the hydrophobic atoms in the side chains of E66 T101 and P107. A mutation to a larger hydrophobic valine side chain at this position displays a destabilizing effect by affecting the packing interactions around this site. The predicted ΔΔG is –0.1 kcal/mol.

The low mutating frequency of NSP10 residues highlights the importance of the role NSP10 plays in the SARS-CoV-2 life cycle. NSP10 is a cofactor that is required for the proper function of two

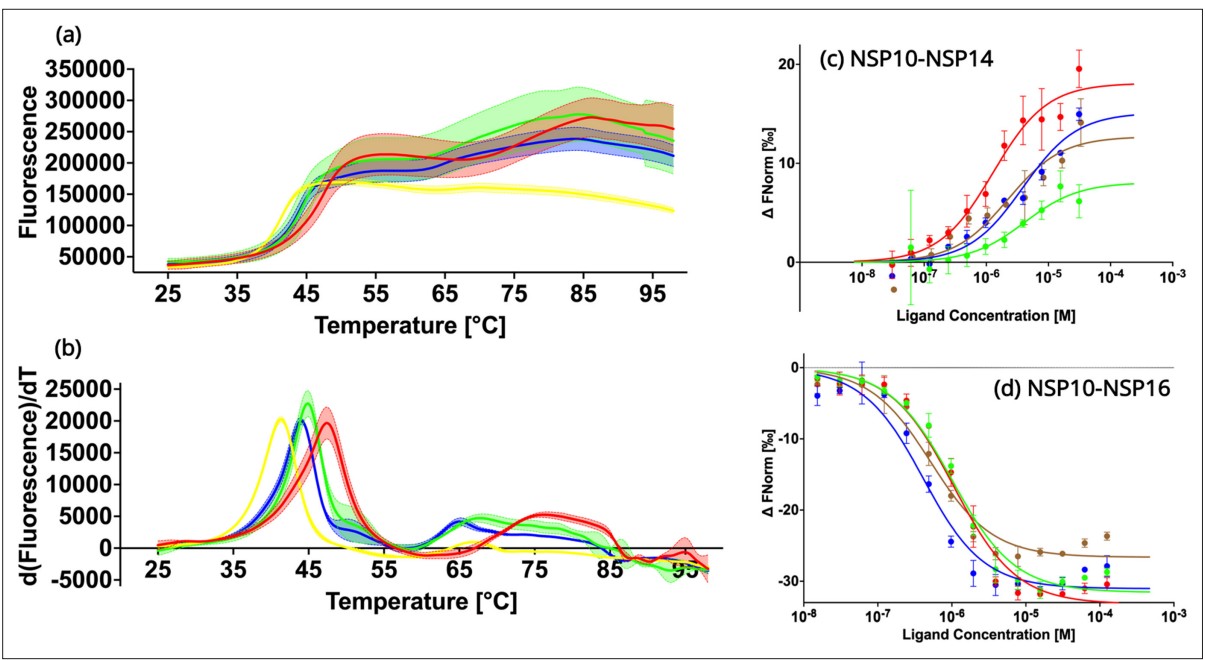

**Figure 3.** Biochemical characterization of native NSP10 and the three most frequent variants. (**a**) Melting curve of native NSP10 and its three variants. The red, yellow, green, and blue curves show the data measured for native NSP10, T12I, T102I, and A104V variants, respectively. (**b**) Derivative curves of NSP10 and its three variants using the same color code. (**c**) MST dose response curves for native NSP10 and its variants in the presence of NSP14 ExoN domain. The green, red, blue and brown curves represent native NSP10, T12I, T102I, and A104V, respectively. (**d**) MST dose response curves for native NSP10 and its variants in the presence of NSP16, with the same color codes as above. All experiments were conducted at least in triplicate.

essential viral enzymes - NSP14 and NSP16. Any significant changes in the structure of NSP10 or the interactions that alter the binding of NSP10 with NSP14 or NSP16 would have deleterious effects on the survival of the virus in the host. Therefore, to adapt, evolve, and self-preserve, the virus ensures that the mutation frequency is low and with negligible effects that threaten its survival.

## Thermal shift assays confirm predicted destabilizing effects in NSP10 variants

In order to experimentally verify the predicted destabilizing effects of the NSP10 variants compared to native NSP10, their thermal stability was tested using Thermal Shift assays (*Figure 3a and b*; *Huynh and Partch, 2015*). The observed $T_M$ values for all three variants T12I, T102I, and A104V were 41.3 ± 0.1°C, 44.0 ± 0.1°C, and 44.9 ± 0.1°C, systematically lower than the $T_M$ value for native NSP10 (47.4 ± 0.0°C). All three variants display a significant change of more than 2 °C, in particular mutant T12I, whose $T_M$ value decreased by 6 °C. Based on the structural comparison, the mutants were predicted to be less stable than the native structure. In summary, these predictions of the stability of NSP10 variants match the experimental data.

## The three frequent NSP10 variants have no significant effect on the binding affinities for NSP14 and NSP16

To quantify the interaction of NSP10 variants with their binding partners, NSP14 and NSP16, MST experiments were conducted (*Figure 3c and d*). With respect to NSP16, for wild-type NSP10, the $K_d$ value was 957.1 nM ±181.5 nM, in a similar range as recently observed for NSP14 (*Kozielski et al., 2022*). Similarly, for T12I, the $K_d$ value was 1000.2 nM ±195.0 nM, which does not show any significant difference compared to native NSP10. For variants T102I and A104V, $K_d$ values were 359.4±97.8 nM and 473.5±117.4 nM, respectively. Their values are smaller than that of wild-type NSP10, indicating a slightly stronger binding to NSP16 (0.38–1.04 fold).

Quantification of the binding affinity between wild-type and variant NSP10s and the ExoN domain of NSP14 MST was employed as previously reported but with a distinct labeling procedure. Instead of using the second-generation labeling dye we covalently attached the fluorescence marker, which allowed freezing and storing the protein at –80 °C the labeled protein for later use. For wild-type NSP10 we measured a $K_d$ of binding to ExoN of 3.9±1.0 µM. For NSP10 variants T12I, T102I, and A104V, the measured $K_d$ values were 1.46±0.38 M, 3.88±1.39 µM and 2.37±1.36 µM, indicating that as observed for NSP16, the variants either show values very close to wild-type NSP10 or display slightly better $K_d$ values (0.37–0.99 fold). In summary, this signifies that the three variants have only marginal effects on the NSP10 binding affinity to NSP14 and NSP16 and show either the same or slightly improved affinities for complex formation. In no case did we observe a drop in affinity.

## Variant T102I does not cause significant structural changes in NSP10

Crystallization experiments were conducted for the three NSP10 variants but only T102I variant yielded protein crystals. The structure was determined by molecular replacement (*Figure 4a and b*) and refined to allow comparison with wild-type NSP10. Data collection and refinement statistics are shown in *Table 2*. Compared with native NSP10, T102I does not display any significant conformational changes. Both structures overlay with an RMSD of 0.26 Å (*Figure 4c*). Even locally, there are no structural changes in residues interacting with the T102I variant (*Figure 4d*). The only observable difference was the loss of a water bridge interaction between the hydroxyl group in the side chain of T102 and a water molecule (*Figure 4d*). This interaction is not present in the T102I mutant. Besides, there are also no structural changes in variant T102I.

## The wild-type and variants display similar conformational dynamics

A detailed comparison of the wild-type structure with that of the T102I variant structure did not reveal any obvious conformational differences. Thus, we performed enhanced sampling molecular dynamics simulations of the wild-type and the three T12I, T102I, and A104V variants. Well-tempered Metadynamics (WT-MetaD) introduces a time-dependent bias potential that is used to influence and accelerate the dynamics to explore conformations that were not previously visited. This ensures a fast spatial and temporal exploration of the Free Energy surface (FES) in the CV space. Comparing the differences in the free energy landscape sampled by the backbone dihedral angles of the wild-type

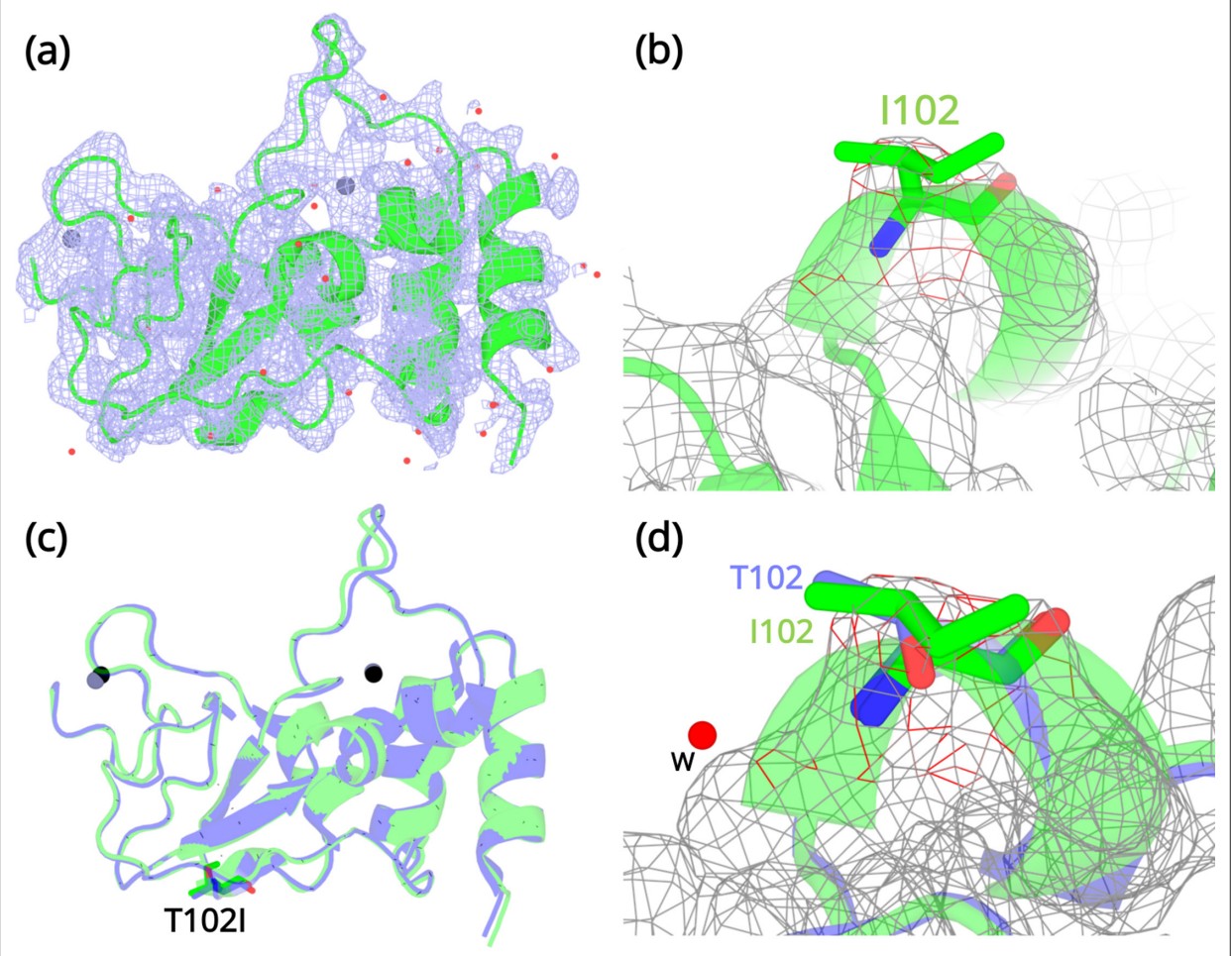

**Figure 4.** Structural details of the NSP10 T102I variant. (**a**) 2Fo-Fc map (1σ) of the NSP10-T102I variant. (**b**) Magnification of the I102 side chain in the electron density map. (**c**) Structural overlay of wild-type NSP10 (PDB entry 6ZCT; blue) and variant T102I (green). The position of T102I is illustrated as sticks (**d**) Overlay of T102 (blue) and I102 (green) side chains. The position of a water molecule present in wild-type NSP10 that makes interactions with the T102 side chain is shown. Zinc atoms are illustrated as grey spheres and water molecules are illustrated as red circles.

and point mutations helps in understanding the structural changes induced by the mutations. The backbone dihedral angles contribute to the slow dynamics in proteins (*Skliros et al., 2012*, *Naritomi and Fuchigami, 2013*). A small number of low-frequency modes are sufficient to describe large fluctuations of a protein and significantly contribute to its conformational change. Besides describing the slow dynamics of a protein, we reasoned that the backbone dihedral angles will be independent of any changes introduced by the varying side chains of the mutations and will also be best suited to describe backbone entropy introduced by a point mutation. The choice of this CV has successfully been used to explore conformational changes in proteins (*Barnes et al., 2018*, *Sangodkar et al., 2017*, *Taylor et al., 2019*).

Six FES plots were generated as a function of and dihedral angles of residues T12, T102, A104, I12, I102, and V104. Next, we compared the wild-type and the corresponding variant FES plots. Ideally, a FES plot represents regions of conformational space of a system. On this free energy map, a potential energy minimum pertains to a particular stable conformational arrangement with relative depths indicative of relative stability or enthalpy. Thus, a protein exploring different metastable conformational states should localize in distinct minima on the FES plot. In this study, the FES plots provide evidence that the wild-type and its corresponding variants explore the same regions on the FE landscape indicating similar conformational dynamics adopted by the backbone dihedral angles. The dihedral angles in the variants do not explore any additional minima than that observed in wild-type NSP10.

**Table 2.** Data collection, data processing, and model refinement statistics for SARS-CoV-2 NSP10 variant T102I.

Data in parenthesis correspond to the highest resolution shell.

| Data collection and refinement statistics | SARS-CoV-2 NSP10 variant T102I(PDB entry 8BZN) |
|---|---|
| Wavelength (Å) | 0.9655 |
| Resolution range [Å] | 42.81–2.19 (2.269–2.19) |
| Space group | I213 |
| Unit cell parameters [Å;°] | a=b=c=104.867, $\alpha=\beta=\gamma=90$ |
| Molecules per asymmetric unit | 1 |
| Total reflections | 23664 (1293) |
| Unique reflections | 9432 (963) |
| Multiplicity | 2.5 (2.6) |
| Completeness [%] | 94.0 (97.8) |
| Mean I/sigma(I) | 11.9 (2.2) |
| Wilson B-factor | 40.45 |
| R-meas [%] | 6.6 (56.9) |
| Rpim [%] | 3.8 (33.6) |
| CC1/2 [%] | 99.7 (72.4) |
| Reflections used in refinement | 9432 (963) |
| Rcryst/Rfree [%] | 17.8 (28.0/23.3 (35.6)) |
| Total no. of non-hydrogen atoms (protein) | 1005 |
| No. of protein / ligand / solvent atoms | 925 / 9 / 71 |
| RMSD bond length, bond angles [Å;°] | 0.012 / 1.04 |
| RamachandranFavored / allowed / outliers / rotamer outliers [%] | 95.9 / 4.1 / 0.0 / 1.0 |
| Clashscore | 4.38 |
| Average B-factor / protein / ligands / solvent | 44.8 / 44.7 / 55.3 / 45.6 |

The most stable conformations were extracted and analyzed from the free energy minima of the wild-type and the variants. The superimposed structures are illustrated in *Figure 5*. The overall root-mean-squared deviation between the wild-type and variants were 0.77 Å (T12I), 0.69 Å (T102I), and 0.63 Å (A104V). Interestingly, we can extrapolate the conformations (denoted by a star) of the wild-type crystal structure (PDB entry 6ZCT) and the T102I crystal structure (PDB entry 8BZN) in the largest minimum of the FE landscape. Altogether, the enhanced sampling simulation data shows that the variants are unable to influence a major conformational change that could lead to altered NSP10 variant binding to NSP14 or NSP16.

## Discussion

Studying and understanding the effects of novel variants of SARS-CoV-2 may provide a plausible explanation for the speed of spread, severity, and impact on immunity of COVID-19. Genome sequencing and alignments on SARS-CoV-2 genomic samples allow identifying variations in the viral genome and more specifically in the various viral proteins (*van Dorp et al., 2020*). The genome sequence data of SARS-CoV-2 is therefore highly valuable in studying the effects of mutations on the function of SARS-CoV-2.

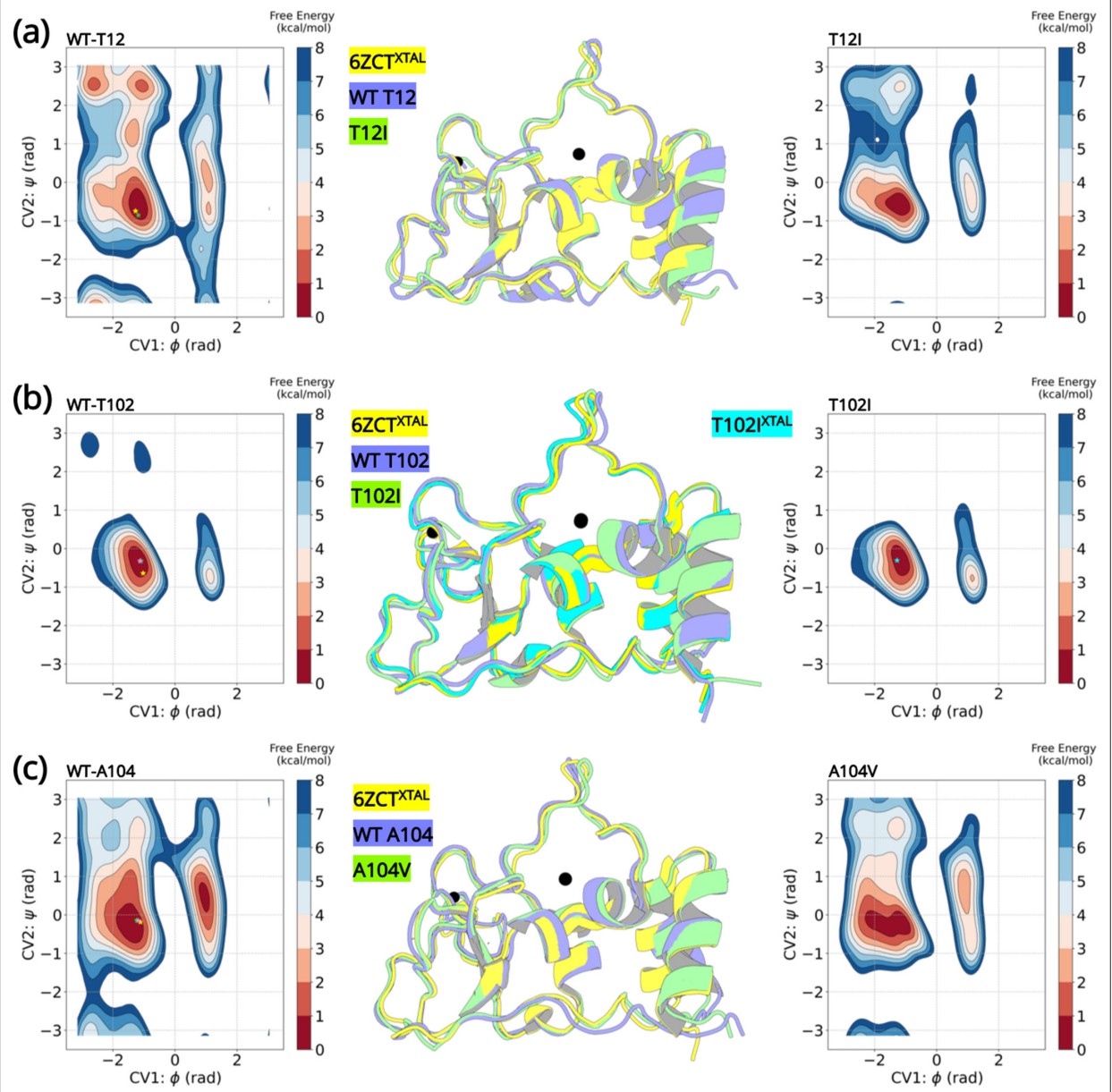

**Figure 5.** Well-tempered metadynamics reveal similar conformational dynamics for the wild-type and variant NSP10. Free energy surface plots of the (**A**) wild-type T12 and T12I variants, (**B**) wild-type T102 and T102I variants and (**C**) wild-type and A104V variants. The wild-type and the variants explore similar conformational landscapes. An overlay of the structures extracted from the largest populated minima highlights similar conformations. The crystal structure 6ZCT$^{XTAL}$ (yellow), the representative conformation extracted from the largest minima in the wild-type simulation (slate blue), and the mutant simulation (green) are superimposed. The resolved crystal structure conformations are extrapolated on the minima and illustrated as a star (yellow- 6ZCT$^{NSP10}$; green - 7MC5$^{NSP14}$; pink - 6W4H$^{NSP16}$; cyan - T102I$^{XTAL}$).

The results from the mutation screening indicate that all residues in NSP10 mutate, albeit with hugely different frequencies. There are only six unique positions in NSP10 that mutate, with an observed frequency greater than 0.1% (*Figure 2b*). The propensity of mutations for various SARS-CoV-2 genes (over the course of COVID19 pandemic) was further calculated using the changes in the number of amino acids observed in the variants of concern (VOC) as a reflection of functionally or adaptively successful mutations using the equation:

$$Mutational\ Propensity = \frac{sum\ of\ total\ observed\ amino\ acid\ changes\ in\ VOC}{Length\ of\ protein}$$

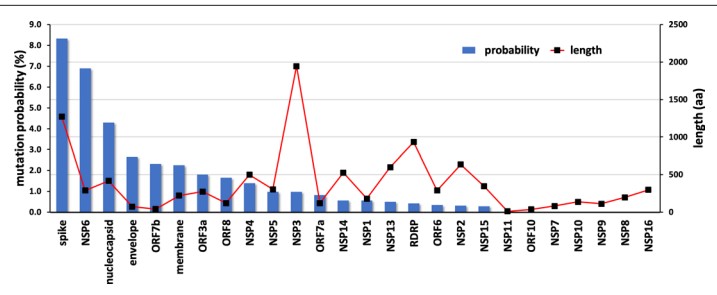

**Figure 6.** Mutational propensities of various SARS-CoV-2 genes. The number of mutations observed in each gene at the protein sequence level and normalized by sequence length is depicted (as blue bars). The genes are ranked in order of their mutational propensity.

The propensities for all SARS-CoV-2 genes are shown in *Figure 6*. The results strongly suggest that NSP10, along with other cofactor proteins, have to date shown a lower propensity to mutation compared to the rest of the proteins. Consider spike protein for instance, 99,254 unique protein sequences have been identified in 2,111,175 quality controlled genomic sequences of SARS-CoV-2 available in the NCBI's Genbank as of December 2022 (https://pandemics.okstate.edu/covid19/#methods). Similarly, for NSP3 115,373 unique protein sequences have been identified in 2,371,934 quality controlled genomic sequences, whereas for NSP10 only 735 unique protein sequences have been identified in 3,070,170 quality controlled genomic sequences. A plausible explanation of this could be because NSP10 acts as an essential cofactor and stimulator of NSP14 ExoN and NSP16 2'O-MTase activities. Any mutation that results in a structural change of NSP10 may alter its binding affinity to NSP14 and NSP16 and thereby affect their activities. This in turn may have an impact on the replication and survival of SARS-CoV-2. In addition, mutations in essential residues forming the two zinc finger domains in NSP10 may compromise structural integrity and therefore function.

The three most frequent mutations are T12I, T102I, and A104V (*Table 1*). There are no frequent mutations that occurred around the two zinc cluster sites in NSP10. More importantly, all identified mutations are also outside the protein-protein interface of NSP10-NSP16 complexes. In contrast, there are eight mutations including P8S, T12I, S33N, G34R, H48Y, T58I, I81T, and K95R, in NSP10 that are at the protein-protein interface of the NSP10-NSP14 complex. Of these mutations, only T12I shows a frequency greater than 0.1%. This suggests that the other mutations in NSP10 may not significantly affect the stability and functionality of the NSP10-NSP14 and NSP10-NSP16 complexes. In spite of the three mutations T12I, T102I, and A104V exhibiting a slightly destabilizing effect, they were predicted to not be detrimental to the function of NSP10.

Although mutations occur throughout NSP10, our analysis reveals that those happen at a very low frequency, compared to other SARS-CoV-2 proteins such as Spike or NSP6, with fewer than 20 times. This implies that multiple mutations are unlikely to happen on NSP10 and that the integrity of this important scaffolding protein and stimulator is essential for SARS-CoV-2, in contrast to other non-structural proteins, which allow for more mutations.

Having been able to quantify the propensity of SARS-CoV-2 NSP10 residues to mutate and create new variants, we are now in a position to apply this knowledge to the recently identified fragment binding pockets using x-ray crystallography (*Kozielski et al., 2022*). In a previous study, we have reported two ligand binding sites (*Kozielski et al., 2022*). Each site involves a distinct set of residues in the interaction with fragments. In addition, both sites involve a symmetry mate to form a binding pocket. Interestingly, both ligand binding sites are located close to the protein-protein interface of the NSP10-NSP14 and NSP10-NSP16 complexes with an overlapping binding interface on NSP10 (*Figure 2d*). Viruses are capable of developing mechanisms of resistance to overcome drug treatment and one mechanism is through the generation of mutations or variants (*Harvey et al., 2021*). For each of the binding pockets, we set out the probability to overcome the effects of ligand binding to these pockets. Residues involved in forming ligand binding site 1 involve S11, T12, and S15. Whereas S11 and S15 show a low mutation frequency T12 displays the highest frequency of all NSP10 residues (*Figure 2—figure supplement 2*). The situation is similar for residues in binding pocket 2. The majority of residues forming this site show medium mutation frequency with the exception of T49,

which shows high frequency (*Figure 2—figure supplement 2*). In conclusion, we hypothesize that certain residues forming the two known ligand binding pockets have the capacity to mutate, which may confer resistance to ligand binding in these pockets.

During our experimental work on the three most frequent variants, we showed that the variants in NSP10 did not lead to major changes in affinity for either NSP14 or NSP16. In the best case they only slightly improved affinity between the binding partners. We posit that a higher affinity of variants for their binding partners may lead to a slightly increased capping activity. However, this is difficult to prove as stimulation is challenging to access experimentally and has not yet been quantified for NSP16. This would signify that the multi-protein and complex SARS-CoV-2 capping system is well balanced and not affected by the most frequent variants in NSP10. Therefore, we do not consider this slight increase in affinity as a significant change.

In conclusion, we disclosed a full list of variants in SARS-CoV-2 NSP10 based on more than 13 million genomic sequences and characterized the three most frequent variants experimentally. We show that these variants do not have profound effects on NSP10 structure and function. We also analyzed amino acids forming recently reported ligand binding sites in NSP10 for their capacity to mutate, which may have an impact on targeting these sites for drug development.

## Materials and methods
### Materials
10 kDa centrifuge filters were purchased from Sigma-Aldrich. Five ml HisTrap FF crude columns were bought from GE Healthcare. NuPAGE MES SDS Running Buffer (20 X), SeeBlue Plus2 Pre-stained Protein Standard, SimpleBlueTM SafeStain, NuPAGE Sample Reducing Agent (10 X), NuPAGE LDS Sample Buffer (4 X) and NuPAGENovex 4–12% Bis-Tris Protein Gels (1.0 mm 10 well) were obtained from Life Technologies. Quick StartTM Bradford 1 x Dye Reagent was from Bio-Raid. SnakeSkin Dialysis Tubing was purchased from Thermo Scientific. Linbro plates were bought from Hampton Research. NT. 115 premium capillaries were obtained from Nanotemper. NSP10 expression clones for mutants T12I, T102I, and A104V optimized for *E. coli* expression was purchased from Genscript.

### Methods
#### Sequence retrieval
Global Initiative Sharing All Influenza Data (GISAID) is an online repository that stores up-to-date genome sequences (https://www.gisaid.org) (*Elbe and Buckland-Merrett, 2017*, *Khare et al., 2021*, *Shu and McCauley, 2017*). In addition to Influenza virus data, GISAID also provides virological and epidemiological information regarding β-coronaviruses.(36) In this study, all genome sequences for SARS-CoV-2 were downloaded in FASTA format (Supplementary data). The sequences for NSP10 were then retrieved from the SARS-CoV-2 dataset and saved separately. The sequences for NSP10 with unknown residues (named X) were removed. Further analyses such as sequence alignment, mutation screening, and mutation effect studies were carried out on the remaining sequences. A workflow of the sequence retrieval and alignment is illustrated in *Figure 2—figure supplement 3*.

#### Sequence alignment
The sequence alignment between all WGS sequences and the Wuhan/WIV04/2019 reference genome was carried out in two phases. We first performed the sequence alignment using the CoVsurver application provided by GISAID (accessed on January 30, 2022 and totaled 7,070,539). The CoVsurver displayed the result of genome sequence alignments by listing all amino acid mutations at every location in the protein. In-house Python scripts were used to quantify this large dataset and identify all mutations in NSP10. The frequency in percentage for the top mutations in NSP10 was calculated using the following equation:

$$\% \, frequency = \frac{total \, count \, of \, mutations}{total \, number \, of \, sequences} \times 100$$

The mutations detected in NSP10 were compiled into one Excel sheet (*Figure 2—source data 1*). From 1747 unique sequences, the top three variants were identified as T12I, T102I, and A104V, following which they were submitted for crystallographic structure determination.

Just before the submission of the manuscript, we updated our sequence analysis (phase 2). A total of 13,032,424 sequences were analyzed (accessed on December 14, 2022). From this 2564 unique sequences were identified. The % frequency was reassessed, which again identified T12I, T102I, and A104V to be the highest occurring variants (*Figure 2—source data 2*). In this manuscript, we report on the updated sequence analysis.

## Structural information for SARS-CoV-2 NSP10, NSP14, and NSP16

The protein structures of native NSP10 (PDB entry 6ZCT; *Rogstam et al., 2020*), the NSP10-NSP14 complex (PDB entry 7MC5)(*Moeller et al., 2022*), and the NSP10-NSP16 complex (PDB entry 6W4H) (*Rosas-Lemus et al., 2020*) from SARS-CoV-2 were retrieved from the Protein Data Bank (PDB). The interface residues script (pymolwiki.org/index.php/InterfaceResidues) for Open-source-Pymol (https://github.com/bieniekmateusz/pymol-mdanalysis) was used to identify the interface residues between NSP10 and NSP14 and between NSP10 and NSP16. The identified interface residues are crucial to the binding of NSP10 to either NSP14 or NSP16 to form the NSP10-NSP14 and NSP10-NSP16 complexes (*Bouvet et al., 2014*).

## Structural analysis

All trajectories were aligned to their crystal structure conformation using Moleculekit implemented in HTMD tools (*Doerr et al., 2016*). To visualize the structures representing each state, the structures collected from the PCCA distributions were loaded and superimposed in Pymol-mdanalysis (https://github.com/bieniekmateusz/pymol-mdanalysis). The structural analysis was performed using mdtraj (*McGibbon et al., 2015*) and MDAnalysis (*Michaud-Agrawal et al., 2011*). Figures capturing major conformational changes were generated using the Protein Imager (*Tomasello et al., 2020*). All plots were made using the matplotlib libraries (*Hunter, 2007*).

## Subcloning, expression, and purification of NSP10 and its variants

Native NSP10 was subcloned, expressed, and purified as previously described (*Rogstam et al., 2020*). The expression vectors for variants T12I, T102I, and A104V in vector ppSUMO-2 were generated by introducing individual point mutations in the respective positions in the codon-optimized expression construct of wild-type NSP10, resulting in constructs coding for an N-terminal His-tag, followed by a SUMO tag, a ULP1 protease cleavage site, and the mutated NSP10 cDNA.

NSP10 mutants were expressed and purified as recently described for native NSP10 (*Rogstam et al., 2020*). In brief, the plasmids were transformed into BL21(DE3) Codon +RIPL competent cells. Cultures were grown in TB medium at 37 °C until an $OD_{600}$ of 0.6–0.7 and induced with 0.5 mM isopropyl-β-D-thiogalactopyranoside (IPTG), allowing protein expression at 20 °C for another 22 hr. Cells were harvested by centrifugation at 8000 *g* at 4 °C for 20 min (Avanti J-E centrifuge from Beckman Coulter, rotor: JLA 16.250).

Cell pellets were resuspended in lysis buffer (50 mM Sodium Phosphate pH 8.0, 300 mM NaCl, 20 mM Imidazole, and 1 mM PMSF) and sonicated for 10 rounds (30 s on 60 s off each round) on ice. Cell debris was removed by centrifugation at 20,000 rpm at 4 °C for 1 hr (Avanti J-E centrifuge from Beckman Coulter, rotor: JLA 25.50). The supernatant was collected and loaded onto a pre-equilibrated 5 ml His-Trap FF crude column. After washing the column with 250 ml wash buffer (50 mM Sodium Phosphate pH 8.0, 300 mM NaCl, 20 mM Imidazole), proteins were eluted with 100 ml elution buffer in 5 ml fractions (50 mM Sodium Phosphate, pH 8.0, 300 mM NaCl and 500 mM Imidazole). The elution fractions containing the nsp10 variant were pooled and divided into two protein samples to be treated differently. One protein sample was dialyzed in the final buffer (50 mM Tris pH 8.0 and 150 mM NaCl), concentrated to 50 mg/ml, and flash-frozen in liquid nitrogen before being stored at 80 °C for use in MST experiments. The second protein sample was dialyzed in dialysis buffer (0.1 mM Sodium Phosphate pH 8.0, 2 mM DTT, and 300 mM NaCl) with ULP1 protease overnight at 4 °C before being loaded to a second 5 ml His-Trap FF crude column followed by washing with 100 ml wash buffer. The flow-through and wash fractions containing cleaved NSP10 variants were both collected and pooled. Protein sample was dialyzed in dialysis buffer and then concentrated to 60 mg/ml.

## Expression and purification of SARS-CoV-2 NSP14 and NSP16

The N-terminal ExoN domain (residues 1–289; $NSP14_{1-289}$ of SARS-CoV-2 NSP14 was subcloned, expressed, and purified as previously described.(28) His-tagged ExoN was then pooled and cleaved overnight with ULP1 protease supplemented with 3 mM DTT. Simultaneous with the cleavage, the protein was also dialyzed against buffer containing 50 mM Tris pH 7.5 and 300 mM NaCl overnight. Thereafter, the protein was passed through the His-Trap column for a second time and washed with 50 mM Tris pH 7.5, 300 mM NaCl and 20 mM imidazole. The cleaved protein without His-SUMO tag was captured in the flow-through, which was then pooled and dialyzed for 18 hr in a buffer containing 50 mM HEPES pH 7.5 and 300 mM NaCl. Lastly, the protein was pooled and concentrated to 2 mg/mL, using an Amicon ultraconcentration device (Millipore) and prepared for MST assays. The expression clone of ppSUMO-2_SARS-CoV-2 NSP16 was obtained from Genscript. Expression was carried out in *E. coli* BL21-ArcticExpress (DE3)-RIL competent cells (Agilent Technologies, Santa Clara, CA, USA) in Terrific Broth modified medium (Melford, Chelsworth, UK) supplemented with 50 µg/ml kanamycin and 20 µg/ml gentamycin. Cultures were incubated at 37 °C, 220 rpm until the $OD_{600}$ reached 0.6–1.0. NSP16 expression was induced by adding 1 mM IPTG and incubated at 10 °C, 220 rpm for 24–30 hr. Cells were harvested by centrifugation at 8000 *g*, 4 °C for 20 min. Cell pellets were resuspended in NSP16 buffer A (60 mM HEPES pH 7.0, 300 mM NaCl, 20 mM Imidazole and 1 mM PMSF), flash-frozen in liquid nitrogen, and stored at –80 °C.

Cell pellets were thawed at room temperature and lysed by sonication on ice for 10 cycles (30 s on, 60 s off) with 16 µm amplitude. Cell debris was removed by centrifugation at 50,000 *g*, 4 °C for 1 hr. The supernatant was loaded into a HisTrap FF crude column (Cytiva, Uppsala, Sweden) pre-equilibrated with nsp16 buffer B (60 mM HEPES pH 7.0, 300 mM NaCl and 20 mM Imidazole). The column was washed with 50 CVs NSP16 buffer B, followed by eluting with 20 CVs NSP16 buffer C (60 mM HEPES pH 7.0, 300 mM NaCl, and 250 mM Imidazole). Samples containing NSP16 were pooled and furtherly purified through a HiLoad 16/600 Superdex 200 pg size exclusion column (Cytiva, Uppsala, Sweden) in nsp16 buffer D (60 mM HEPES pH 7.0, 300 mM NaCl and 1 mM DTT). Fractions containing pure nsp16 were pooled, aliquoted, flash-frozen in liquid nitrogen, and stored at –80 °C.

## Protein crystallization

Protein crystallization was performed in Linbro plates using the hanging drop method. All conditions were optimized using published crystallization conditions for native SARS-CoV-2 NSP10 (0.1 M Bis-Tris pH 5.5–6.5 and 1.8 M-2.4 M NaCl)(*Moeller et al., 2022*). Only NSP10 variant T102I formed crystals grown at 20 °C, whereas the other two variants did not yield any crystals, despite various attempts. Crystals were incubated in cryoprotectant solution for 5 min containing 1.2-fold of the reservoir solution supplemented with 20% DMSO, mounted in loops, flash frozen in liquid nitrogen, and stored in pucks in liquid nitrogen.

## X-ray diffraction data collection, structure determination, and refinement

Diffraction data for the NSP10 variant were collected automatically at ID30A-1 beamline at the European Synchrotron Radiation Facility (ESRF, France). For measurements, the beam size was adjusted to 50 µm to cover the entire crystal. A total of 230 images were collected with an oscillation range of 0.10° (23° in total). The data were processed with autoPROC to a resolution of 2.4 Å (*Vonrhein et al., 2011*). The structure was solved by molecular replacement with SARS-CoV-2 NSP10 as a search model (PDB entry 6ZCT) using PHASER-MR of the PHENIX suite (*Adams et al., 2010*). Electron density and difference density maps, all σA-weighted, were inspected, and the model was improved using *Coot* (*Emsley et al., 2010*). The structure was refined with PHENIX.(44) The calculation of $R_{free}$ used 5% of the data. Crystallographic and refinement statistics are provided in *Table 1*. In the NSP10 variant structure, there are no flexible regions, and the structure covers residues T7 to C130 and contains two zinc fingers.

## Thermal shift assays (TSA)

TSA assays were performed to compare the thermal stability of NSP10 variants with native NSP10. NSP10s were diluted to 120 µM using the appropriate buffer (50 mM Tris pH 8.0 and 150 mM NaCl) and mixed 1:1 with 20 x Sypro Orange Dye. Samples were distributed in 96-well PCR plates (40 µL

samples each well) and sealed with transparent sealing film. The measurements were done in triplicate using a 7500 Real-Time PCR System (Thermo Fisher). The samples were heated from 25°C to 99°C at a rate of 1 °C/min. The data were analyzed using Applied Biosystems Protein Thermal Shift Software.

## Determination of the NSP10-NSP16 and NSP10-NSP14 binding affinities using MicroScale Thermophoresis (MST)

His-tagged NSP16 was diluted to 400 nM in NSP16 buffer D and mixed with an equal volume of 100 nM RED-tris-NTA $2^{nd}$ Generation labeling dye (NanoTemper Technologies GmbH, München, Germany). The mixture was incubated on ice for 1 hr and centrifuged at 15,000 $g$, 4 °C for 10 min to remove aggregates. An MST pre-test with 10 nM labeled sample was carried out to determine the labeling and sample quality. To determine the $K_d$ values of native nsp10 and its three variants to nsp16, nsp10s were 2-fold serial diluted from 500 μM for 15 rounds in ligand buffer (50 mM Tris pH 8.0 and 150 mM NaCl). An equal volume of 20 nM labeled NSP16 was mixed with diluted samples and incubated on ice for 15 min prior to the measurements. All measurements were conducted by the Monolith NT.115 instrument using Monolith NT.115 Premium Capillaries. Binding affinity data was collected and measured by the MO.Control and MO.Affinity Analysis software.

Protein labeling was carried out by using NanoTemper Protein Labeling Kit RED-NHS $2^{nd}$ Generation. For NSP14$_{1-289}$, 10 μL of the 300 μM RED-NHS $2^{nd}$ Generation labeling dye and 90 μL of the 10 μM protein sample were mixed carefully and incubated for 30 min at room temperature in the dark. During the incubation, B-column was equilibrated with 10 mL assay buffer (50 mM HEPES pH 7.5 and 300 mM NaCl). After incubation of the labeling reaction, 100 μL of dye-protein solution was transferred to the equilibrated B-column. 550 μL assay buffer was then added to the B-column, to remove excessive free dye and the labeled protein was eluted by adding another 450 μL assay buffer, aliquoted, flash-frozen in liquid nitrogen, and stored at –80 °C for subsequent use.

NSP14$_{1-289}$ was diluted to 20 nM using 1 x PBS-T buffer. Native NSP10 and the three variants were diluted to 500 μM. 20 μL NSP10 was transferred into the first well of a 96-well microplate. NSP10 was then diluted to final concentrations of 5E-1, 2.5E-1, 1.25E-1, 6.25E-2, 3.13E-2, 1.56E-2, 7.81E-3, 3.91E-3, 1.95E-3, 9.77E-4, 4.88E-4, 2.44E-4, 1.22E-4, 6.10E-5, 3.05E-5 and 1.53E-5 mM by serial dilution in 1 x PBS-T buffer. 10 μL of 20 nM NSP14$_{1-289}$ was added into each tube from 16 to 1 and mixed carefully. Monolith NT.115 premium capillaries were dipped into each tube from 1 to 16 and placed in positions 1–16 of the device tray for the measurements. Measurements were done in triplicate by using the Monolith NT.115 instrument (NanoTemper Technologies, Munich, Germany). The binding affinity tests for NSP14 ExoN and each NSP10 variant were repeated by the same procedure as carried out for native NSP10.

## Effects of the variants on the stability of NSP10 and its complexes with NSP14 and NSP16

Thermodynamics plays a critical role in examining the structural stability of proteins. In summary, such examination is achieved by comparing the free energies (ΔG) of the wild-type and variant proteins; given that the two protein states, even of the same species, are likely to have very different free energies. The DynaMut2 algorithm predicts the effects of missense mutations on protein stability and dynamics (*Rodrigues et al., 2021*). It combines normal mode analysis to capture protein motion and graph-based signatures to represent the wild-type environment to assess the impact of point mutations on protein stability and dynamics, for example in our case, between wild-type and variant NSP10. It can accurately predict the effects of NSP10 variants by calculating the vibrational entropy and the Gibbs free energy changes of the variants. We further assessed the effects of single mutations in NSP10 on the protein-protein interactions in complexes using Mutabind2 software (*Zhang et al., 2020*). Mutabind2 predicts the effects of mutations by calculating the changes in binding affinities and binding free energies. Seven features are utilized, including those describing interactions of proteins with the solvent, evolutionary conservation of the site, and thermodynamic stability of the complexes. The output estimates whether the mutation will be detrimental to the complex.

### Molecular dynamics set up

The crystal structure of NSP10 (PDB entry 6ZCT)(*Rogstam et al., 2020*) and NSP10-T102I variant (PDB entry 8BZN) were downloaded from the PDB and used as the starting geometry for wild-type

and variant simulations, respectively. The high-throughput molecular dynamics (HTMD) protocol was employed to prepare the simulation systems (*Doerr et al., 2016*). The NSP10-T12I and NSP10-A104V variants, which we were unsuccessful in crystallizing, were generated from the wild-type by replacing the residues at position 12 and 104 using the HTMD package (*Doerr et al., 2016*). The protonation states for the side chains were calculated using *proteinprepare* implemented in the Moleculekit module (*Martínez-Rosell et al., 2017*). The protonated PDB file was subsequently adapted to the Zinc Amber force field (ZAFF) format by adopting the ZAFF nomenclature that best represented the coordination of the metal center (*Peters et al., 2010*). The system was then placed in a TIP3P waterbox whose edges were at least 10 Å from the solute atom (*Price and Brooks, 2004*) The systems were minimized with 5000 steps of steepest descent and equilibrated in the NPT ensemble for 5 ns. The Langevin thermostat and Berendsen barostat were used to keep the temperature at 300 K and pressure at 1 bar (*Berendsen et al., 1984*). Two independent simulations were run as replicates for each system, which included wild-type NSP10s (T12, T102 and A104) and the variant-type (T12I, T102I, and A104V). All the simulations were run using the ACEMD v3.5 MD engine (*Harvey et al., 2009*). The simulation protocol was identical for all simulations.

### Free energy calculations using well-tempered metadynamics simulations

Well-tempered metadynamics (WT-MetaD) simulations were run to study the influence of point mutations on finite sampling (*Bonomi et al., 2009*, *Laio and Gervasio, 2008*). The $\varphi$ and $\psi$ dihedral angles of the wild-type and point mutations (residues T12, T102, A104) were chosen as the collective variables for enhanced sampling simulations. The choice of the CVs was based on the observation that the slowest motions in a protein are a function of their backbone flexibility (*Naritomi and Fuchigami, 2013*, *Skliros et al., 2012*). The structural effects, resulting from the differences between the interactions of the wild-type and point mutation side chains should be observable in the dihedral angles.

PLUMED 2.7 metadynamics code implemented in ACEMD version 3.5 was used to run WT-MetaD simulations (*Harvey et al., 2009*). The bias factor was set at 15 kT. The initial Gaussian height was set to 0.5 kJ/mol, deposited every 4 ps, so that the deposition rate was equal to 0.125 kJ/(mol.ps). The Gaussian width was set to 0.1 rad for the two CVs. A total of 5 µs in the NVT ensemble was needed to reach full convergence of the free energy. The sampling convergence was checked by comparing the reconstructed free energy surfaces at different time intervals. The values of the CVs were stored in the COLVAR file and the Gaussians were saved in the HILLS file. A free energy surface (FES) was constructed as a function of the two CVs, by integrating the deposited energy bias along the trajectory. The conformations from each minimum were retrieved using clustering based on Cα RMSD with a cutoff of 2 Å.

## Acknowledgements

We acknowledge the European Synchrotron Radiation Facility for the provision of synchrotron radiation facilities and we would like to thank Dr. Matthew Bowler for assistance in using beamline ID30A-1 (MASSIF-1) to measure T102I variant data. We are grateful to the MRC - UCL Therapeutic Acceleration Support (TAS) for financial support. This publication contains part of the doctoral thesis of HW, DD, JL, QW, and WS. FK and SH are funded by Medical Research Council grant MR/X013995/1.

## Additional information

### Competing interests
Shozeb Haider: Reviewing editor, *eLife*. The other authors declare that no competing interests exist.

### Funding

| Funder | Grant reference number | Author |
| --- | --- | --- |
| Medical Research Council | MR/X013995/1 | Frank Kozielski<br>Shozeb Haider |

| Funder | Grant reference number | Author |
|---|---|---|

The funders had no role in study design, data collection and interpretation, or the decision to submit the work for publication.

## Author contributions

Huan Wang, Formal analysis, Investigation, Writing - original draft; Syed RA Rizvi, Danni Dong, Formal analysis; Jiaqi Lou, Formal analysis, Investigation; Qian Wang, Watanyoo Sopipong, Investigation; Yufeng Su, Fares Najar, Software, Investigation; Pratul K Agarwal, Supervision, Investigation, Writing - review and editing; Frank Kozielski, Supervision, Funding acquisition, Investigation, Methodology, Project administration, Writing - review and editing; Shozeb Haider, Conceptualization, Data curation, Funding acquisition, Validation, Visualization, Methodology, Project administration, Writing - review and editing

## Author ORCIDs

Pratul K Agarwal ⬚ http://orcid.org/0000-0002-3848-9492
Shozeb Haider ⬚ http://orcid.org/0000-0003-2650-2925

Joint Public Review: https://doi.org/10.7554/eLife.87884.3.sa1
Author Response https://doi.org/10.7554/eLife.87884.3.sa2

# Additional files

## Supplementary files

• MDAR checklist

## Data availability

All files required to run the simulations (topology, coordinate, plumed input), processed trajectories (xtc), corresponding coordinates (pdb), COLVAR and HILLS files for each system described in this manuscript can be downloaded from https://doi.org/10.5281/zenodo.7477127.

The following dataset was generated:

| Author(s) | Year | Dataset title | Dataset URL | Database and Identifier |
|---|---|---|---|---|
| Wang H, Rizvi SRA, Dong D, Lou J, Wang Q, Sopipong W, Su Y, Najar F, Agarwal PK, Kozielski F, Haider S | 2023 | Emerging variants of SARS-CoV-2 NSP10 highlight strong and functional conservation of its binding to two non-structural protein, NSP14 and NSP16 | https://doi.org/10.5281/zenodo.7477127 | Zenodo, 10.5281/zenodo.7477127 |

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
