## [Editor Report · eLife assessment]

This study presents an **important** discovery that the RNA synthesis protein of SARS-CoV-2, the virus that is responsible for COVID 19, has fewer mutations and causes limited conformational changes. The evidence supporting the claims is **convincing**, with robust sequence alignment studies, state-of-the-art protein-protein interaction analysis, and molecular conformational analysis. This work has implications for drug design and will be of broad interest to the general biophysics and structural biology community.

---

## [Referee Report · Joint Public Review]

The work is of fundamental importance and is a useful structural resource to the SARS-CoV2 proteome. The work relies on large-scale SARS-CoV2 genomes and extracts frequent mutations in two key proteins NSP16 and NSP10. The impact of these mutations was studied using x-ray crystallogrpahy, biophysical assays and simulations to propose structural changes. The evidence is, therefore, convincing to suggest NSP10 conformational changes are limited. More analysis on functional implications would be useful to understand the underlying reasons of limited structural variability. The questions raised during the review of the original submission have been addressed by the authors.

---

## [Author Response]

The following is the authors’ response to the original reviews.

**Replies to reviewers**

For instance, The DynaMut2 and thermal shift assays point towards less stable variants than wild type, with Tm values slightly lower. On the other hand, the Kd value of variants reported stronger binding of NSP10 with NSP16. How do authors explain this, as the change due to point mutation may not fall within error range?

Concerning the lower Tm values for the mutants compared to wild type NSP10, the errors of the measurements conducted in triplicate are very low (0.1 degrees) indicating that they do not fall into the error range, in particular as the changes in Tm are significant with changes of up to 4 degrees. This is consistent with the DynaMut23 calculations. Furthermore, the differences in Kd values between wild type and mutants are partially significant. Whereas one of the mutants did not display any changes in Kd value. Compared to wild-type NSP10 for both NSP14 and NSP16, the other show a 2 to 3 fold better Kd, with reasonable errors and we consider those as small but significant, and not within error range.

For instance, the conformational ensemble could be utilized for docking with NSP16 and NSP14. There could be a potential alternative pathway for explaining the above changes in Kd. This should be attempted for understanding the role in its functional activity.

We agree with the reviewer. We are working on a follow up manuscript exclusively looking into the NSP10-NSP14/16 interfacial interactions. Our preliminary results from biophysical and biochemical analysis suggests a range of Kd values observed between the mutants and the NSP14/NSP16. We are also investigating changes in the interfacial interactions via crystallography.

Therefore, more quantitative analysis is required to explain structural changes. The free energy landscape reported in the paper may not capture rare transition events or slight rearrangements in side chain dynamics, both these could offer better understanding of mutations.

We agree with the point raised by the reviewer. As mentioned above, we are exclusively looking into these interfacial interactions and binding between different partners, which will be reported in a follow up manuscript.

Recommendations for the authors: please note that you control which, if any, revisions, to undertake

1. Line 206, V104 need to be corrected to A104.

done

2. Line333, does it mean the Kd value of NSP10 binding to NSP16 similar to the Kd value of binding to NSP14?

Yes. Overall, they are in about the same range with a Kd value of around 1 µM for the NSP10-NSP16 complex and 4 µM for the NSP10-NSP14 complex.

3. Figure 3, the colors corresponding to different variants or native NSP10 could be consistent for easier reading and understanding.

The colors have been edited.

4. The data presented in Figure 3d are not clear enough to draw conclusions about the Kd Value in the main text.（Values of variants are smaller than that of wild-type NSP10, indicating a slightly stronger binding to NSP16）

The measured differences are small with 2 to 3 fold differences, but significant and are not within the error range as can be derived from the data and calculated Kd values and their errors.

5. Are there other mutations in the sequence with the top 3 mutations? If yes, is it possible to do the same experiments with that protein? Why not choose the NSP10 of the popular strain for the determination of the binding ability to NSP14 and NSP16.

No, the top three were single point mutations.

6. Enzyme activity assays like ExoN activity detection of NSP14 and vitro activity detection of NSP16 2′-O-MTase could be performed to characterize the effect of these three mutations on biological function.

Yes, it would be good to consider these. We are considering these assays in the follow up manuscript as mentioned above.

7. More details on image acquisition and writing errors need to be clarified and corrected.

Done.

8. Typo in Results section T12, T102, V104 should be A104

Done.

9. DynaMut analysis is extrapolated to explain that "Mutation to a hydrophobic side chain such as Ile, results in a loss of this interaction." There is no data to support this as complexes have not been studied. Perhaps this is speculative at best.

We have changed this sentence to “Mutation to a hydrophobic side chain such as Ile, is predicted to result in the loss of this interaction”, since this was a prediction